# The Use of Cryotherapy in Cosmetology and the Influence of Cryogenic Temperatures on Selected Skin Parameters—A Review of the Literature

**Adrianna Dzidek [1] and Anna Piotrowska [2],***

1    Doctoral School of Physical Culture Science, University of Physical Education, 31-571 Krakow, Poland
2    Institute for Basic Sciences, Faculty of Physiotherapy, University of Physical Education,
     31-571 Krakow, Poland
*    Correspondence: anna.piotrowska@awf.krakow.pl

**Abstract:** Cryotherapy is becoming an increasingly popular method used in medicine, physiotherapy, and cosmetology. It is used in the form of whole-body cryotherapy (WBC) and local cryotherapy. It is a tool for achieving analgesic and anti-inflammatory effects. Since the beginning of its use, its influence on the mental state has also been pointed out. The aim of this study was to analyze the available literature on the effect of cryogenic temperatures on the skin and the mechanisms induced by such a stimulus and its influence on well-being. A literature search of keywords or phrases was performed in PubMed®. Various effects of WBC on skin characteristics (hydration, pH, level of transepidermal water loss), mechanisms of anti-inflammatory effects, and effects on adipocytes were shown. Research on the impact of individual skin characteristics is not consistent. Positive effects on the reduction of inflammation and oxidative stress have been noted. Cryotherapy is also successfully used in dermatology to treat lentil spots, actinic keratosis, and ingrown toenails, remove viral warts, or relieve itching in atopic dermatitis. The results of the review also indicate the effectiveness of WBC as an adjunctive treatment for obesity. The number of papers available on the direct effects of WBC on the skin is still limited, despite the fact that it represents the first contact of the human body with cryogenic temperatures. Available data show that cold as a physical stimulus can be a safe and useful tool in cosmetology.

**Keywords:** whole-body cryotherapy; local cryotherapy; skin hydration; transepidermal water loss; skin pH

## 1. Background

Whole-body cryotherapy (WBC) is defined as the stimulus transmission of cryogenic temperatures (below $-100\ °C$) for a short period of time (1–3 min) to the surface of the whole body in order to induce physiological responses to cold. By directly affecting the entire body, cold as a stimulus used in WBC affects the endocrine and nervous systems [1]. The aim of cold therapy is to induce physiological or psychological benefits [2]. The purpose of cryotherapy is not to cool the target tissue, but to bring heat from the warmer tissue to the tissue with a lower temperature [3].

Local cryotherapy occurs through the direct contact of cold with the skin, through evaporation or convection. The action of direct contact is the use of cold, cooling or ice compresses, ice massages, and immersion in cold and icy water. Evaporation is a method of generating cold on the skin by means of a rapidly evaporating liquid spray that uses the heat of vaporization. Convective cryotherapy is the introduction of nitrogen gas ($-160$ to $-180\ °C$) or cold air ($-20$ to $-30\ °C$) to the skin surface. Deliberate internal administration of cooled $CO_2$ ($-50$ to $-70\ °C$) using a specialized probe causes necrosis of the nerve fibers and thus alleviates pain. This method is used for faceted arthrosis [4].

Another form of cold therapy is outdoor swimming in icy waters (rivers, pools, or lakes). Research suggests that winter water baths bring about a number of benefits to the body: Improving endocrine and hematological functions, alleviating mood disorders, and improving overall well-being [5]. People who regularly use such water baths report improved immunity. Stimulation with cold has also been found to improve the ability to cope with stress [6].

Single exposures, as well as the use of a series of cryotherapeutic treatments, are subject to many scientific studies. WBC therapy has been and is used, first of all, in diseases of the musculoskeletal system (e.g., osteoporosis, post-traumatic and overload changes in joints and soft tissues, fibromyalgia, and inflammatory diseases) and the nervous system (e.g., multiple sclerosis, spastic paresis) [7,8]. The positive impact of the WBC on mental health has been demonstrated: It is used to alleviate depression and anxiety syndromes [9]. WBC is also used in sports medicine. It was shown that extreme cold can improve metabolism, physical strength, and act as a pain reliever and improve recovery after exercise [10,11]. It is known that the interaction between the action of cold and the body occurs mainly through the skin. The impact of cold treatments on the condition and functioning of this tissue is still being understood. The aim of this study was to analyze the available literature on the effect of cryogenic temperatures on the skin and the mechanisms induced by such a stimulus and its influence on well-being.

The term cryotherapy is used to describe the treatment of patients. If, however, the procedure is to stimulate or support bioregeneration, the term cryostimulation is used.

Nowadays, it is recognized that procedures in the cryochamber induce a number of beneficial physiological reactions, such as analgesic [10], anti-edema [12], and anti-inflammatory [13] effects, which are associated with effects on the neuromuscular [14], vascular [15], hormonal [1], and immunological [6] systems. Cryotherapy applied locally is used as a form of treatment, treatment support and rehabilitation, and as a pain-relieving factor in some procedures. Other forms of local application of cryogenic temperatures are cryoablation and cryosurgery.

## 2. Materials and Methods

The work is a narrative review. The PUBMED database was searched using keywords to indicate the relevant literature items published up until 2021. The terms allowing the search for relevant works were the following keywords and their combinations: Whole body cryotherapy, local cryotherapy, cryostimulation and skin, oxidative stress, inflammation, skin properties, and fatty tissue. The analyzed scientific items were of both research and review natures. The data were evaluated to draw conclusions that would answer the research problem. The article that did not meet the criteria were excluded. The quality assessment was performed by both researchers, independently. A total of 7 articles focused on the analgesic effect (3 studies, 4 reviews), 6 on inflammatory markers (5 studies, 1 review), 12 on oxidative stress (5 studies, 7 reviews), 25 on the impact on the skin (15 studies, 10 reviews), 8 on use in dermatology (3 studies, 5 reviews), and 9 on the influence on body fat (5 studies, 4 reviews).

## 3. Results

### 3.1. Analgesic Effect

The analgesic effect of cryogenic temperatures is due to the combined effect of several mechanisms. Cryostimulation affects the nervous system (switching off sensory receptors and their connections with proprioreceptors using cold and slowing down the conductivity in sensory fibers) and the hormonal system (increasing the concentration of beta endorphins) [16]. Pertovaara et al. [17] proved experimentally that spinal administration of noradrenaline (NA) to animals relieves pain, which is associated with the activation of the sympathetic nervous system. Huttunen et al. [18] noted that winter bathing significantly increased the level of NA in the plasma. Thus, adrenergic stimulation and the constriction of blood vessels during and after exposure to cold may have a significant effect on reducing the perception of pain. Another mechanism is the influence on the release of inflammatory

mediators, which is seen in the case of inflammatory pain [11]. On the other hand, data from Algafly et al. [19] suggest that cryotherapy may increase the pain threshold and pain tolerance, which is associated with a significant decrease in nerve conduction velocity. In their studies, the pain threshold and pain tolerance were assessed by a pressure algometer (Pain Diagnostic and Thermography, Great Neck, NY, USA) at two different sites. The first assessment site was iced and the second one was non-iced. The circular probe head of the algometer was lowered gradually at a steady rate until discomfort was reported (pain threshold) and removed at the point of the pain becoming unbearable (pain tolerance). The study showed that with a decrease in skin temperature, an increase in pain threshold and pain tolerance occurred [19].

Systemic and local cryostimulation is used to delay muscle soreness after physical effort, relieve pain in sports injuries, and assist in the treatment of joint pain, lower back pain, tendinitis, or sprains [11,20]. In this type of study, the most common assessment tool is the Visual Analogue Scale (VAS).

Cold therapy is increasingly used in beauty salons. The effect of cold on the nerve terminals and the nerve conduction velocity will produce an anesthetic effect [19]. Therefore, it can be used before, during, or after a painful procedure such as laser therapy. This technique is also used to improve the comfort of permanent makeup: It reduces redness and possible bleeding of the treated areas [21]. Different studies on the effectiveness of this method in offsetting pain sensations have not yet been conducted, and it seems that all kinds of subjective tools will be sufficient for evaluation.

### 3.2. Influence of Cryotherapy on the Level of Inflammatory Markers

Inflammation is a set of processes aimed at defending the body against an inflammatory initiating agent (e.g., microbial, chemical, or physical). It is characterized by the presence of an increased internal temperature, increased blood flow, redness, pain, and impaired function of the inflamed tissue. If the inflammation continues chronically, even at a low intensity, it will contribute to degradative changes, aging of the body, and the appearance of a number of pathophysiological changes including those affecting the skin.

It has been noted that exposure of the body to extreme cold reduces the intensity of inflammatory processes; fewer pro-inflammatory mediators are released, and more anti-inflammatory cytokines are released [13]. Under the influence of low temperatures, the concentration of the acute-phase protein CRP (c-reactive protein), erythrocyte sedimentation rate (ESR), and the concentration of IgA and IgG immunoglobulins and pro-inflammatory interleukins (IL-2, IL-8) decrease. On the other hand, the concentration of interleukin-10 (IL-10), which has an anti-inflammatory effect, increases. In addition, cryotherapy improves humoral and cellular immunity, stimulating B lymphocytes and NK lymphocytes (natural killers) [22]. Inflammation is characterized by increased plasma concentrations of circulating proinflammatory cytokines (IL-6, IL-9, and TNF-$\alpha$, among others). Ziemann et al. [23] demonstrated that the application of a series of 10 WBC treatments reduces TNF-$\alpha$ concentrations, indicating an anti-inflammatory effect of cryostimulation. Pilch et al. [24] noted that a series of 20 cryostimulation treatments significantly reduced CRP protein values among obese subjects, while no significant changes were noted in subjects with normal body mass. This is interesting because obesity is a recognized pathophysiological factor causing chronic inflammation, which increases the likelihood of developing further disease entities of the metabolic syndrome. This also demonstrates that cryochamber treatments will not decrease the concentrations of selected cytokines per se, but only normalize their concentrations.

The activation of inducible nitric oxide synthase (iNOS) during inflammation in macrophages or muscle cells is thought to result in the release of large amounts of nitric oxide (NO). It is well known that NO is a major vasodilatory mediator and also inhibits the expression of proinflammatory cytokines [25]. On the other hand, NO may be a trigger for inflammation. Więcek et al. [26] subjected 20 men (mean age of 59) to a series of 24 WBC treatments, at a frequency of every other day. In their study, they demonstrated that a series of WBC treatments would increase the expression of iNOS, regardless of physical activity level. Więcek et al. [26], simultaneously with iNOS, examined the levels of CRP and interleukins, namely IL-6, IL-10, and IL-1β. They noted no changes in inflammatory markers. This shows that the body's responses regarding inflammatory markers after exposure to cryogenic temperatures may be different. For clinical conditions that proceed with an increase in inflammatory markers, WBC allows them to decrease. For organisms with physiological concentrations, the treatments will have no effect.

### 3.3. Cryotherapy and Oxidative Stress and Skin

The theory of aging regarding the effect of reactive oxygen species (ROS) was proposed in 1956 by Denham Harman. He suggested that free radicals accumulate over time and are one of the major factors that contribute to the aging of the body [27]. This concept was the basis for the development of the free radical theory of mitochondrial aging (MFRTA) [28]. ROS are thought to be formed by electron leakage in the respiratory chain in mitochondria. This causes damage to the cytosolic components and the mitochondria themselves. A further consequence is the aging of the body. This theory assumed that ROS are the only source of mitochondrial damage, and it is impossible to block their formation and thus completely repair mitochondria and other subcellular components [29]. However, recently, this theory in relation to the whole organism has been questioned [30–32]. In spite of that, the free radical theory of aging holds true for the second largest human organ, the skin. There is a clear correlation between exposure to ROS from both extrinsic and intrinsic sources and the pro-aging effect [33]. Oxidative stress during aging reduces the expression of antioxidant enzymes, their concentration, or activity, including SOD (superoxide dismutase), CAT (catalase), and GPx (glutathione peroxidase). In addition, the total oxidative status [TOS] increases, while the antioxidant capacity (TAC) decreases [34].

The influence of WBC on the antioxidant status and activity of antioxidant enzymes was studied in healthy subjects, athletes, and patients with various disorders. Miller et al. [35] noted that a series of 10 WBC treatments resulted in an increase in total antioxidant status (TAS) in patients with multiple sclerosis. Wojciak et al. [34] demonstrated an increase in the GPx activity in erythrocyte and plasma TAC levels after the first WBC treatment in a group of older, physically active men. They also showed an increase in CAT activity after a single treatment in young, non-training individuals. Pilch et al. [36] observed a significant increase in CAT activity after a series of 20 WBC treatments in subjects with obesity. Sutkowy et al. [37], in a study involving young, healthy men, demonstrated an increase in the SOD and the GPx activity after a single WBC treatment. Using a series of treatments (at least 12 WBC treatments) resulted in an increase in SOD activity, and increasing the number of treatments to 24 potentiated this effect [36]. The SOD activity was also shown to increase after a series of 20 WBC treatments [38].

### 3.4. The Impact of Cryotherapy on Skin

The skin is responsible for many defense, secretory, and sensory functions [39]. It is designed to protect the human body against external harmful factors, at the same time ensuring contact with the external environment and recognizing its stimuli. In addition, the skin is equipped with immune cells and a number of receptors essential for the host's defense and the maintenance of tissue homeostasis [40].

The human body is homeothermic, which ensures its proper functioning. In turn, the external temperature (measured on the skin) is dependent on the environment. The maintenance of relative thermal homeostasis is ensured by adipose tissue and hair, which do not allow excessive heat loss.

Skin temperature can be measured using a digital thermometer. An electrode is applied directly to the skin, so temperature measures reflect skin changes with heat lost rather than the direct effect of cold on the thermometer [19]. Another tool for assessing skin temperature is the Hanna Instruments HI-8751 telethermometer with an HI-765Ped factory-calibrated (ISO 9000) thermistor surface probe [41]. Cholewka et al. in their studies used thermography and contact thermometry. Thermography was performed with a Thermovision Camera AGEMA Type 470 and a Thermovision Camera A40 (Flir Systems Company, Danderyd, Sweden). The distance between the camera and the body needs to be approximately 1.0–1.5 m, depending on the height and size of the human. Contact thermometry was performed using thermocouples Ni-Cr-Ni-Al stacked to the surface of the body with a hypoallergic plaster [42].

In WBC, the temperature of the skin follows the low ambient temperature [43]. Exposing the skin to extreme cold causes significant fluctuations in the skin temperature, which leads to an increase in the activity of skin thermoreceptors and, consequently, to the stimulation of the thermoregulation center in the hypothalamus [14,44]. It was indicated that changes in skin temperature closely correlate with changes in subcutaneous and intramuscular temperatures [41,44]. However, Jutte et al. [45] showed that the variability of skin temperature is only responsible for 21% of the variability of the temperature in the muscle. In this research, the absolute value of temperatures was shown, therefore it cannot be directly compared with results in other papers. Nevertheless, the results of the studies by Jutte et al. [45] show that the relationship between skin temperature and the temperature of deep tissues is not clear.

The reaction to cold has two stages. In the first stage, sympathetic adrenergic fibers are stimulated, releasing NA. The speed of nerve conduction slows down and local blood vessels narrow [14,46]. In the second stage, there is a rapid expansion of blood vessels and tissue reperfusion. Such stimulation of microcirculation with increased reperfusion after limited blood flow in the narrowed vessels increases the supply of oxygen to cells and other nutrients, which will likely also improve the appearance of the skin. Such an action is to be expected with repeated exposure to low temperatures. It is indicated that the stimulation of microcirculation will support the penetration and distribution of medicinal or cosmetic preparations used immediately after the WBC treatment [1]. As indicated earlier, WBC has an inhibitory effect on the amount of ROS and stimulates the number of antioxidants produced by the body [34,35,37]. Therefore, the use of multiple cryotherapy treatments may bring effects similar to those observed after the use of cosmetics or nutricosmetics that improve the prooxidative-antioxidant balance [1,47].

With age, an accumulation of cellular damage leading to metabolic dysfunction and chronic inflammation is observed [48]. Aging cells are responsible for the appearance of wrinkles and morphological changes in elastin fibers [49]. They are unable to divide but remain metabolically active. They secrete numerous pro-inflammatory cytokines (including IL-6, IL-8), growth factors, chemokines, and proteases, known as the aging-associated secretory phenotype (SASP). It is shown that the number of SASP-expressing fibroblasts increases with age [50,51]. It has been also noted that in keratinocytes of aging skin, the secretion of IL-1$\alpha$ is higher, which may be related to the inflammation process occurring in these cells [52]. It contributes to a malfunction of whole skin tissue. The pro-inflammatory environment of the skin affects the reconstruction and damage of the extracellular matrix (ECM), thus disrupting the wound-healing process. Therefore, especially in the elderly, the inflammation process will affect the appearance and architecture of the skin and also the immune function of the skin [50].

**Table 1.** Changes in selected skin features due to single or repeated whole-body cryotherapy treatments.

|  | Subjects | Intervention | Observation |
|---|---|---|---|
| Piotrowska et al. [53] | women (*n* = 43) and men (*n* = 33) 23.63 (SD = 1.36) | 1 | Hydration: no impact TEWL: deterioration pH: no impact |
| Skrzek et al. [54] | women (*n* = 20) 5.87 (SD = 7.54) | 10 | Hydration: improvement after the first 3 treatments, deterioration after the entire series pH: no impact Lubrication: no impact |
| Kang [55] | women (11) 30–40 | 12 | Hydration: improvement Lubrication: improvement |

SD: standard deviation; TEWL: Transepidermal Water Loss.

The number of available studies describing the direct impact of cryotherapy on skin parameters is very limited (Table 1). Piotrowska et al. [53] showed the effect of a single systemic cryotherapy treatment on the improvement of skin hydration. As a result of high temperatures, the sweat glands secreted large amounts of fluids and electrolytes. Shamsuddin et al. [56] proved that the ion reabsorption capacity of the sweat gland is significantly lower at cool ambient temperatures compared to optimal. Therefore, it can be presumed that the eccrine glands, in response to extreme cold, will produce sweat slower, which will reduce water leakage from the extracellular space and, consequently, will improve the hydration of the stratum corneum. Skrzek et al. [54] also noted an improvement in the level of hydration after a one-time WBC treatment, but only in three out of eight measured locations (the chin contour, right cheek, and right lower limb). After applying a series of 10 treatments, Skrzek et al. [54] showed a decrease in the level of skin hydration in all measured locations, except for one (the forehead). This exception of the forehead may be due to a large amount of sebum secreted in this area. A significant difference in the cited works is the age of the participants and the number of treatments. Piotrowska et al. [53] investigated the effect of a one-time WBC procedure on skin features among women and men, whose mean age was 23.63 years. Skrzek et al. [54] studied middle-aged women (a mean age of 58.7) who underwent a single WBC treatment and a series of 10 treatments. With age, decreased vasoconstriction in the skin was observed during exposure to cold. Changes in skin blood flow reactions related to aging may contribute to thermoregulation disorders [57], which may have an impact on the results of the level of hydration in the respondents in the studies presented above. Therefore, the initial improvement in skin hydration demonstrated by Piotrowska et al. [53] could, as in the case of Skrzek et al. [54], be reversed under the influence of a series of treatments. However, this hypothesis would require further research to confirm it.

Kang [55] also noted an improvement in epidermal hydration in his research. After a series of treatments, the level of hydration in the group subjected to cryotherapy was 58% and was 53% in the control group. In his research, Kang [55] also noted the effect of cold treatments on the amount of sebum on the skin: 34% in the study group, while in the control group, the change was 27%. The authors indicated the presence of a clear correlation between the level of hydration and the degree of sebum secretion. In line with this observation, it can be concluded that in the study by Skrzek et al. [54], the level of sebum should decrease with the decrease in the level of hydration. However, the authors of the studies did not notice a decrease in sebum levels as a result of cryotherapy [54]. It may be related to the age of the probes and the activity of the sebaceous glands depending on the activity of the endocrine system [58].

The pH value of the skin is assessed by the presence of water on the surface of the skin. Changes in the skin microcirculation and the activity of sweat glands generated by cryogenic temperature will modulate the amount of water on the skin surface, which may imply changes in the acid–base reaction. Both Piotrowska et al. [53] and Skrzek et al. [54] suggest lowering the skin pH under the influence of WBC, but their results were not statistically significant. It

is worth noting that, in both studies, the pH value was within the limits considered beneficial for the skin (in Piotrowska et al. [53], 4.0–6.00; in Skrzek et al. [54], 5.5–6.08).

### 3.5. Cryotherapy in Dermatology

Dermatological cryotherapy currently uses temperatures ranging from –70 °C to −196 °C, depending on the area of indication. In dermatology, cryotherapy includes any type of application of freezing techniques to remodel, elevate, or destroy diseased tissue or its components [59].

Local cryotherapy using liquid nitrogen is used in dermatology as a therapeutic method. It is used to treat lip injuries, acne scars, and keloids [60], and is also a technique for exfoliating the epidermis (cryopeeling) in the case of photo-damaged skin. Due to the fact that melanocytes are destroyed already at 7 °C, this procedure is also recommended in the case of melanosis [61].

Cryotherapy is also one of the treatments used to treat lentil spots [21] and actinic keratosis (AK) [62], which is a major precursor to squamous cell carcinoma. Liquid nitrogen is also used to remove viral warts caused by the human papilloma virus (HPV) [63].

Cooling the skin relieves itching in atopic dermatitis (AD). Sanders et al. [64], in their studies on the AD mouse model, noted that heating the skin increases scratching attacks and significantly shortens the onset of scratching. They observed an inverse relationship in the case of skin cooling—the onset of scratching was delayed, and the number of attacks decreased.

Cryotherapy can be also used to treat ingrown toenails. Moreover, Turan et al. [65] confirmed in their retrospective study that it can greatly and consistently improve the dermatological life quality index in suitable ingrown toenails.

### 3.6. WBC vs. Body Fat

Exposure of the body to extreme cold requires high synchronization of the endocrine and circulatory systems and the activation of a number of metabolic processes in order to ensure the body's homeostasis. The somatic and autonomic nervous system is also stimulated, which results in increased catecholamine secretion and stimulation of β-adrenergic receptors. These processes intensify, among others, ATP synthesis, lipolysis, and glycogenesis and increase the activity of membrane transport in muscles [66].

Fontana et al. [67] reviewed the literature of studies on the effects of WBC on obese patients. The results of the review indicate the effectiveness of WBC as an adjunctive treatment for obesity. The authors point to the potent antioxidant and anti-inflammatory activities of cryostimulation, and the importance of WBC as adjunctive therapy in reducing abdominal obesity and body weight. However, the data shown in the review indicate only the effect of WBC as an adjunct, not as the main agent.

The dynamics of the temperature of the outer surface of the human body depend on the ability of tissues to eliminate the loss of heat transferred from the inside of the body to the skin and the thermal gradient between the external environment and the body. The first factor takes into account many variables, such as the amount of body fat and blood supply to the skin and muscles, which will contribute to the formation of thermal resistance. As the thermal resistance increases, so does the insulating capacity. Tissue cooling depends on sexual dimorphism, which results from the differences in the amount and distribution of adipose tissue and the water content in the body. Consequently, the isolating capacity of men and women may differ [68], which was confirmed experimentally by Cuttel et al. [69]. These authors, in their studies, noticed that women were characterized by lower temperatures assessed on the skin after a one-time WBC treatment than men. This was also confirmed by Polidori et al. [70]. They noticed that the skin temperature in women after the WBC procedure was lower by 2.6 °C in relation to the skin temperature in men. This indicates that women are more sensitive to the effects of extreme cold. According to the authors, the key element of thermal resistance seems to be fatty tissue, related to the percentage of fat [70]. This is confirmed by studies conducted in people with different

body compositions. Wyrostek et al. [71] noted that the factor causing differences in the adaptation to cryogenic temperatures may be the amount of adipose tissue, and therefore, for people with disturbed body composition, the number of cryotherapy treatments should be modified. In addition, in a literature review, Fontana et al. [67] indicated that WBC mimics the effect of exercise-induced WBC, and therefore its efficacy will be related to initial physical capacity and percent body fat.

Cryotherapy as a tool for reducing adipose tissue and maintaining the correct body composition can be considered in two ways: As local treatments in the form of cryolipolysis, using local cooling for non-invasive destruction of adipocytes [72], and systemic, affecting the entire patient's body, including metabolites of adipose tissue (lipokines) [24].

Cryolipolysis is based on the assumption that adipocytes are less resistant to cooling than other water-rich skin cells. Cryolipolysis is a non-invasive method used to reduce the number of localized fat deposits by inducing local inflammation of adipose tissue caused by a low temperature. Proper use of low temperatures causes their apoptosis. In turn, the fat cells are absorbed by macrophages and digested by them [73].

## 4. Conclusions

Cryotherapy has a strong effect on the human body, including the skin, and this effect depends on the number of treatments. The effect of a single visit to the cryochamber may be the opposite of the effect of 10 or 12 treatments, and such series are most often used in rehabilitation or therapy.

Numerous studies indicate a reduction in the severity of inflammation and oxidative stress, which may indirectly affect the aging process of the skin and improve its appearance. An important effect of the treatment is the improvement of blood circulation. At present, there are only a few studies available on the effects of WBC on basic skin characteristics, and the results are not consistent. Nevertheless, the authors unanimously emphasize the safety of cryotherapy treatments for the skin. Further studies on the effects of cryotherapy on the skin seem necessary, and the basic question is how the amount of body fat will modify the effect of cryogenic temperatures on skin tissue in women and men.

### *Study Limitations*

The main limitation of the present review is the small number of publications addressing the issue under study, which directly examined, in an objective manner, the effects of WBC treatments on the skin. A second important limitation is an inability to double-blind the studies. Another limitation is the different treatment protocols and variable conditions for performing treatments in different centers, which implies different degrees of cryogenic effects on the skin.

**Author Contributions:** Conceptualization, A.P.; methodology, A.P.; software, A.D.; validation, A.D. and A.P.; formal analysis, A.D.; investigation, A.D. and A.P.; resources, A.D.; data curation, A.D.; writing—original draft preparation, A.D. and A.P.; writing—review and editing, A.D. and A.P.; visualization, A.D.; supervision, A.P.; project administration, A.P. All authors have read and agreed to the published version of the manuscript.

**Funding:** This research received no external funding.

**Institutional Review Board Statement:** Not applicable.

**Informed Consent Statement:** Not applicable.

**Data Availability Statement:** Not applicable.

**Conflicts of Interest:** The authors declare no conflict of interest.

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
