# Peer review of "The Use of Cryotherapy in Cosmetology and the Influence of Cryogenic Temperatures on Selected Skin Parameters—A Review of the Literature"

_cosmetics, doi:10.3390/cosmetics9050100_

Round 1

Reviewer 1 Report

Main challenge is the lack of consistent scientific studies. It is impossible to do double blind studies, so we will probably never get solid studies studies.

However the findings cited in the review do suggest, that cryotherapy is not just a fad.

As such it is worthy of publication

Author Response

Answers to Reviewer

              Thank You very much for the reviews we received on our article. The work of the Reviewers helped to improve the manuscript, for which we are grateful.

              Below, we will try to provide explanations for the deficiencies or inaccuracies indicated by the reviewers.

Reviewer 1.

Main challenge is the lack of consistent scientific studies. It is impossible to do double blind studies, so we will probably never get solid studies studies.

A new chapter was added:

“Study limitations

The main limitation of the present review is the small number of publications ad-dressing the issue under study, which directly examined, in objective manner, the ef-fects of WBC treatments on the skin. A second important limitation is the inability to double-blind the studies. Another limitation is the different treatment protocols and variable conditions for performin treatments in different centers, which implies dif-ferent effect of cryogenic stimuli on the skin.”

Reviewer 2 Report

Thank you for your sending the manuscript entitled " The use of cryotherapy in cosmetology and the influence of cryogenic
temperatures on selected skin parameters - a review of the literature" for reviewing.  My comments are as follows: [Abstract] Although this is a review article, the abstract can be  formated from Purpose, Material & Method, Results & Discussion, and Conclusion. [Background] The paragraph of "Cryotherapy and Cryostimulation" can be merged with background. [Materials & Methods] The details of the database from PUBMED should be addressed more clear.  The categorizations related to cosmetics.  For example, how many papers related to analgesic effects, inflammatory markers, oxidative stress, impact on skin etc. [Results]  Should have more information in each report and category.  For exasmple;  1. How to measure the pain score?.  2. How to measure the changes of temperature during and after crotherapy.  3. Analysis of the factors such as patient's gender, age, body weight, duration of WBC, time interval of WBC treatment, and nember of WBC treatment.  4. The information of "Cryotherapy in Dermatology" is not enough to fit for the goal of this journal " Cosmetics". [Conclusion]  Since cryotherapy and cryositmulation is the main goal of this study, any solid parameters for the researchers to follow?   In conclusion, this is a very good review regarding the cryotherpy. However, it need more revision brfore acceptance.

Author Response

Answers to Reviewer

              Thank You very much for the reviews we received on our article. The work of the Reviewers helped to improve the manuscript, for which we are grateful.

              Below, we will try to provide explanations for the deficiencies or inaccuracies indicated by the reviewers.

Reviewer 2.

 [Abstract] Although this is a review article, the abstract can be  formated from Purpose, Material & Method, Results & Discussion, and Conclusion.

The abstract has been changed. At this point, we tried to follow the journal's guidelines and take into account the reviewer's suggestions.

 [Background] The paragraph of "Cryotherapy and Cryostimulation" can be merged with background.

As suggested by the reviewer, the indicated fragment was moved to the chapter: Background.

[Materials & Methods] The details of the database from PUBMED should be addressed more clear.  The categorizations related to cosmetics.  For example, how many papers related to analgesic effects, inflammatory markers, oxidative stress, impact on skin etc.

The chapter: Materials & Methods has been updated.

[Results]  Should have more information in each report and category.  For exasmple;  1. How to measure the pain score?.  2. How to measure the changes of temperature during and after crotherapy.  3. Analysis of the factors such as patient's gender, age, body weight, duration of WBC, time interval of WBC treatment, and nember of WBC treatment.  4. The information of "Cryotherapy in Dermatology" is not enough to fit for the goal of this journal " Cosmetics".

The chapter: Results has been supplemented. The changes that were made allowed the character of the work to be maintained: a narrative review. The fragment on Dermatology is not extensive, but it should be pointed out that the discussion of the use of cryotherapy in dermatology was not the direct aim of this paper, we just wanted to highlight this topic. This is due to the dichotomy of dermatology and cosmetology as two specialties with different correspondences

[Conclusion]  Since cryotherapy and cryositmulation is the main goal of this study, any solid parameters for the researchers to follow?  

Chapter Conclusions has not been changed. Our results show that the cryochamber treatments are safe and effective across a broad spectrum of skin welfare mechanisms. This is the most brief and based on these conclusions, we want to encourage more scientists to do a deeper analysis.